# The Role of Cognitive Skills, Sex, and Parental Education for Social–Emotional Skills: A Cross-Sectional Study on the WPPSI-IV Performances of Children Aged 3 to 5 Years

**DOI:** 10.3390/children9050730

**Published:** 2022-05-17

**Authors:** Franziska Walter, Monika Daseking, Franz Pauls

**Affiliations:** 1Department of Medicine, Medical School Hamburg, 20457 Hamburg, Germany; 2Department of Educational Psychology, Helmut-Schmidt-University/University of the Federal Armed Forces, 22043 Hamburg, Germany; m.daseking@hsu-hh.de; 3Department of Clinical Psychology, Helmut-Schmidt-University/University of the Federal Armed Forces, 22043 Hamburg, Germany; paulsf@hsu-hh.de

**Keywords:** WPPSI-IV, preschoolers, social–emotional development, cognition, visual–spatial skills, sex differences

## Abstract

Background: Current research suggests that knowledge about the relationship between cognition and social–emotional skills in preschoolers is important to better understand child development. The present study investigated possible effects of cognitive skills measured by the Wechsler Primary and Preschool Scale—Fourth Edition (WPPSI-IV), children’s sex, and parental educational level on social–-emotional skills measured by the Developmental Test 6 Months to 6 Years—Revision (ET 6-6 R) for children aged 3 to 5. Methods: Statistical analyses were based on a sample of *N* = 93 children (47 females, 46 males). First, bivariate correlations among relevant WPPSI-IV index scores, the ET social–emotional quotient, children’s sex, and parental educational level were calculated to identify possible significant associations between the variables under investigation. Subsequently, two multiple regression analyses were conducted to test for the hypothesized main effects of cognitive skills, children’s sex, and parental educational level on social–emotional skills. Finally, a moderated multiple regression analysis was carried out to investigate whether possible effects of cognitive skills on social–emotional skills were moderated by children’s sex and parental educational level. Results: Regression analyses indicated that visual–spatial skills measured by the WPPSI-IV and children’s sex have both a small but significant main effect on social–emotional skills. The main effect of sex was due to the fact that, on average, females achieved higher scores on the measure of social–emotional skills than males. Conclusions: The present findings suggest that the WPPSI-IV represents a suitable test battery for the assessment of those cognitive skills, which might play a reasonable role in social–emotional development

## 1. Introduction

Recent research has emphasized that knowledge about the relationship between social–emotional skills and different cognitive processes in preschoolers may provide comprehensive insight into the developmental characteristics of young children [1,2]. Since research on this topic has predominantly focused on social–emotional skills and cognitive processes independent of each other, little is still known about the associations among these domains. Thus, most studies on cognition and social–emotion skills have examined one or the other domain, while often neglecting the relationship between cognition and social–emotional skills as well as its overall impact on child development. Nevertheless, it can be suggested that broader skills associated with emotion and cognition are closely related: whereas some work together, others work in opposite directions [3].

Beside the lack of research on the relationship between cognitive skills and social–emotional development, another issue relates to the lack of comparability between studies on this topic. This is mainly due to the fact that most of those studies used different instruments to measure a variety of different cognitive skills. Even though complex test batteries such as the Wechsler scales provide a reliable assessment of different cognitive skills (e.g., *Verbal Comprehension*, *Processing Speed*, *Visual Spatial*, *Fluid Reasoning*, and *Working Memory*), there are hardly any studies that have examined whether specific cognitive skills measured by well-established test batteries are significantly associated with social–emotional skills. Therefore, the current study aims to clarify whether different cognitive skills measured by the German Wechsler Preschool and Primary Scale of Intelligence—Fourth Edition [4] may have a meaningfully high predictive value for social–emotional skills in children aged 3 to 5.

### 1.1. Social–Emotional Skills and Cognitive Skills

Various definitions for social–emotional skills have already been proposed by several researchers, but in general, most of these definitions refer to abilities to recognize and manage emotions, to appreciate the perspectives of others, to constructively handle interpersonal conflicts, to make responsible decisions, and to form positive relationships. For instance, Denham et al. [5] defined social–emotional development as multidimensional in nature and proposed five dimensions to be crucial for it: (1) social competence; (2) attachment; (3) emotional competence; (4) self-perceived competence; and (5) temperament/personality. The authors also emphasized that social and emotional competence are closely linked to social–emotional development. Especially for the preschool period through kindergarten for ages 3 to 5/6 years, Denham et al. [5] defined the following important milestones for social skills: e.g., the beginning of peer interactions while managing emotional arousal, the beginning of specific friendships and peer status, and the emerging of prosocial behaviours and interactions as well as for emotional skills: e.g., the expression of ‘blended ‘emotions, understanding expressions and situations of basic emotions, and more independent emotion regulation skills. It has to be noted that although the majority of those milestones for social as well as emotional skills that have previously been defined by Denham et al. [5] are measured by the test battery used to assess social–emotional development in the present study, the operationalization represents a culmination of the milestones to be focused on. The definition proposed by Denham et al. [5] describes common milestones for the period of kindergarten for children aged 3 to 5 years, which also accurately represents the age group under investigation and therefore strengthens the link between the authors’ theoretical foundation and the measurement within the present study.

The definition of cognitive skills in the present study is based on a body of research focusing on widely accepted structural intelligence models and investigating their factorial validity. These studies indicated strong evidence for a hierarchical model including a general intelligence factor at a superordinate level and specific interrelated but distinguishable broad abilities at a subordinate level. At the lowest level, specific cognitive abilities are each composed of different narrow abilities [6,7]. Most research on structural models indicated that verbal comprehension, visual–spatial skills, fluid reasoning, working memory, and processing speed are among the most important subcomponents of intelligence [6,7].

Research investigating the connection between cognitive and social–emotional skills mainly refers to associations between executive functions (e.g., inhabitation as well as working memory), emotional understanding and regulation, and theory of mind. However, the test battery used in the present study measures executive functions (e.g., working memory) to a lesser degree and does not operationalize crucial aspects of components of theory of mind. Therefore, the following sections focus on those domains of cognitive and social–emotional skills that are operationalized by the instruments used in the present study.

### 1.2. Social–Emotional Skills and Verbal Cognitive Skills

To date, there are several studies indicating that language skills may be a strong predictor for social competence in typically developing children [2,8,9]. Longobardi et al. [10], for instance, were able to demonstrate that language skills may have a direct effect on social competence in a sample of children in an early period of infancy (18 to 35 months), which is a critical phase for lexical development [11]. In their study, three different indices were used to operationalize language skills: the size of expressive vocabulary, measures indicating verbal as well as grammatical and syntactic development, and a measure indicating social aspects of language competence. All of these indices measuring language skills were found to substantially predict measures of social competence. A possible explanation for this findings is that children who speak competently could also be perceived as more socially competent by their social environment. Language might thus help children with scheduling and conducting everyday social activities as they gain a deeper understanding of other’s emotions through talking about it. This connection could then likely strengthen social–emotional skills [12].

Furthermore, research has indicated that children aged 1 to 3 years (toddler age), who suffer from language disorders, including difficulties in both receptive and expressive language or in one of those domains, tend to show considerable impairments in employing social skills, regulating their emotions and behaviour, and demonstrating empathy [13]. Holdnack, Goldstein, and Drozdick [14] investigated social perception and performances on the Wechsler Adult Intelligence Scale—Fourth Edition (WAIS-IV [15]) in a sample of adolescents and adults aged 16 to 40 years and diagnosed with either Asperger’s Syndrome or autism. The study suggested that disorders in the autism spectrum are strongly associated with deficits in social interaction, whereas autism may be more strongly accompanied by additional cognitive deficits than Asperger’s Syndrome. Hence, only the significant differences between the group of Asperger´s syndrome and healthy control group are described in the following sections of the present study. Even though subjects of the autism spectrum group achieved significantly lower scores in the WAIS-IV subtest *Comprehension* and Social Perception than healthy controls, significant group differences were neither found in the WAIS-IV primary index *Verbal Comprehension* nor in other verbal subtests.

### 1.3. Social–Emotional Skills and Non-Verbal Cognitive Skills

In addition to measures of verbal cognitive skills, the WPPSI-IV also provides measures of non-verbal cognitive skills, including *Visual Spatial*, *Fluid Reasoning*, *Working Memory*, and *Processing Speed*. However, index scores for *Fluid Reasoning* and *Processing Speed* can only be calculated for children aged 4 to 7 years. Therefore, the following sections will explicitly focus on the relationship between *Visual Spatial*, *Working Memory,* and social–emotional development.

Working memory has long been known as part of a complex cognitive system supplying limited capacity for temporal information storage. Working memory capacity plays an important role in solving complex cognitive tasks related to reasoning, decision-making, language comprehension, and learning [16]. Research has already indicated that working memory capacity may be mainly responsible for planning and conducting social–emotional behaviour [2]. For example, De Wilde, Koot, and van Lier [17] found that lower working memory capacity is most likely associated with lower levels of likeability by peers, more teacher–child conflicts, and lower levels of teacher–child warmth. These variables were analysed over a 2-year period in a sample of children from kindergarten to elementary school.

Other studies investigated the relationship between working memory and social skills. Barkley [18] suggested that deficits in working memory may affect children’s attention to social norms as well as their understanding of cause and effect. Children often tend to apply skills related to working memory in order to take another individual’s perspective when communicating to significant others. This strategy may then help understanding the opposite and applying social rules [19]. Such a relationship is also supported by Holdnack et al. [14], who demonstrated that subjects diagnosed with Asperger’s Syndrome showed significantly lower performances on WAIS-IV subtests measuring auditory working memory than healthy controls.

Cummins, Peck, and Dyck [20] examined the link between emotional difficulties and deficits in social skills in children with poor motor skills. The authors hypothesized that low empathic skills might be associated with deficits in visual–spatial processing. Other studies indicated that children with developmental coordination disorders may show a broad range of perpetual deficits, including impairments in visual–spatial skills [21] as measured by the subtests *Block Design* and *Object Assembly* of different Wechsler Scales. Deficits in visual–spatial skills are also associated with impaired emotion recognition skills and difficulties in recognizing emotional facial expressions. That means that children featuring such deficits are likely to have problems with the perception and decoding of emotional facial expressions [22]. This, in turn, might then lead to further deficits in social skills.

### 1.4. Children’s Sex, Parental Education, and Social–Emotional Skills

As outlined before, there is some empirical evidence for positive associations between social competence and verbal cognitive skills in typically developing children. However, only a few studies investigated whether this relationship varies depending on sex and research on this topic is quite inconsistent so far. Some studies indicated that associations between social competence and verbal cognitive skills tend to be stronger for boys than for girls when comparing children aged 4 years and older [23,24]. However, research is quite limited when it comes to providing evidence for such a moderation effect of sex in children aged 3 years and younger. Longobardi, Spataro, Frigerio, and Rescorla [25] examined the role of sex differences for the relationship between social competence and verbal cognitive skills in preschoolers aged 18 to 35 months. Whereas their analyses indicated significant and moderately high correlations among measures of social competence and verbal cognitive skills for boys, such correlations could not be found for girls. Stowe et al. [24] suggested that developmental problems in girls at preschool age could be less apparent in boys at about the same age. The results of their study indicated that boys at preschool age with language deficits showed more difficulties in social–emotional tasks than girls. Thus, such problems might be more likely to be perceivable in boys, because they tend to need more social attention and show more disruptive behaviour than girls with language deficits.

Since several studies have also suggested a considerable effect of the parental educational background on both cognitive [26,27] and social–emotional [28,29] skills, thus indicating that a high parental education level encourages both the cognitive and the social–emotional development, parental educational level was additionally included in the analyses of the present study.

### 1.5. Research Questions

As previously described, there is only a small body of research available that provides rigorous investigations on the relationship between different cognitive and social–emotional skills. Therefore, it is undeniable that more research is needed in order to expand our knowledge about whether certain cognitive skills may affect social–emotional skills in children aged 3 to 5 years in particular and may thus contribute to social–emotional development in general. Given that the possible effects of cognitive dysfunctions on social–emotional skills could easily be overlooked due to methodological flaws, the German WPPSI-IV was used as a well-established test battery for measuring the performances on a variety of different cognitive tasks in the present study. This way, rigorous analyses of WPPSI-IV test results should help identify those effects, if existing, and might finally provide important clinical implications as well as improve clinical decision-making. In conclusion, it has to be noted that even though information obtained by comprehensive intelligence tests only is not sufficient for diagnosing specific cognitive disorders, reliable indicators of cognitive skills, such as the primary and secondary indices provided by the WPPSI-IV, may help to detect even small variations in cognitive measures and clarify how those are associated with variations in measures of social–emotional skills.

On the basis of these considerations, the present study aims to address the following research questions:Do WPPSI-IV test performances on *Verbal Comprehension*, *Visual Spatial*, *Working Memory*, and *Vocabulary Acquisition* have a significant effect on social–emotional skills in children aged 3 and 5 years?Do children’s sex and parental educational level have a significant effect on social–emotional skills in children aged 3 and 5 years?Does children’s sex moderate possible effects of *Verbal Comprehension* and *Vocabulary Acquisition* on social–emotional skills in children aged 3 and 5 years?Does parental educational level moderate possible effects of *Verbal Comprehension*, *Visual Spatial*, *Working Memory*, and *Vocabulary Acquisition* on social–emotional skills in children aged 3 and 5 years?

## 2. Materials and Methods

### 2.1. Sample

Data of a sample of *N* = 93 children (47 females, 46 males) were partially gathered from their parents and subjected to all subsequent analyses of the present study (see Table 1 for an overview). One parent per child completed a standardized questionnaire soliciting information about their child’s social–emotional skills as well as a questionnaire assessing socio-demographic variables such as sex, age, parental education, and migration background. Children were recruited from different kindergartens in Bremen and Lower Saxony, Germany. Prior to data collection and testing, parents and kindergarten teachers were informed about the main goals and procedures of the study as well as about data processing and protection. Parents were then required to give their approval by signing a written informed consent form.

### 2.2. Measure

#### 2.2.1. Cognitive Skills (WPPSI-IV)

The WPPSI-IV [4] represents a test of cognitive skills developed for children aged 2 years and 6 months to 7 years and 7 months. In general, two WPPSI-IV test versions are available: a short version for children aged 2 years and 6 months to 3 years and 11 months (age group 2:6–3:11) and a long version for children aged 4 years to 7 years and 7 months (age group 4:0–7:7).

For age group 2:6–3:11, the WPPSI-IV is designed to measure the *Full Scale IQ* (FSIQ) and three primary indices: *Verbal Comprehension Index* (VCI), *Visual Spatial Index* (VSI), and *Working Memory Index* (WMI) as well as three ancillary indices: *Vocabulary Acquisition Index* (VAI), *Nonverbal Index* (NVI), and *General Ability Index* (GAI). These WPPSI-IV indices are calculated based on a total of seven subtests, among which are *Information* (IN), *Receptive Vocabulary* (RV), *Picture Naming* (PN), *Block Design* (BD), *Object Assembly* (OA), *Picture Memory* (PM), and *Zoo Locations* (ZL).

For age group 4:0–7:7, the WPPSI-IV framework comprises a total of 15 subtests. In addition to those seven subtests applied in age group 2:6–3:11, the following eight subtests are included as well: *Similarities* (SI), *Vocabulary* (VO), *Comprehension* (CO), *Matrix Reasoning* (MR), *Picture Concepts* (PC), *Bug Search* (BS), *Cancellation* (CA), and *Animal Coding* (AC). The long version of the WPPSI-IV provides measurements of the *FSIQ* and the following five primary indices: *Verbal Comprehension Index* (VCI), *Visual Spatial Index* (VSI), *Fluid Reasoning Index* (FRI), *Working Memory Index* (WMI), and *Processing Speed Index* (PSI). Ancillary indices can also be calculated for *Vocabulary Acquisition Index* (VAI), *Nonverbal Index* (NVI), *General Ability Index* (GAI), and *Cognitive Proficiency Index* (CPI).

For the German adaptation of the WPPSI-IV [30], the primary indices VCI, VSI, and WMI as well as the ancillary index VAI have indicated excellent internal consistency coefficients across all ages, with Cronbach’s alpha ranging from 0.87 to 0.94.

#### 2.2.2. Social–Emotional Skills (Developmental Test for Children Aged 6 Months to 6 Years—Revision; ET-6-6-R)

The ET-6-6-R [31] is a developmental test for children aged 6 months to 6 years, which includes the following developmental domains: *gross motor skills*, *fine motor skills*, *language development*, *cognitive development*, and *social–emotional development*. The scale for social–emotional development is the only one that is assessed using an external questionnaire for parents; data for all other scales or domains are directly gathered from the child. Item selection was originally based on empirical studies on the development-related boundary stones [32]. Accordingly, the ET-6-6-R provides different combinations of questions for assessing social–emotional development in different age groups (age groups 36–42 months, 42–48 months, 48–60 months, and 60–72 months). While those items that explicitly describe a child’s behavioural characteristics are adapted to the according age group, the majority of items are invariant across the four different versions of the questionnaire. Table 2 provides an overview of some item examples for the age group 60–72 months. Scale scores included in the ET-6-6-R range from 1 to 19 points with a mean of 10 and a standard deviation of 3. The score for social–emotional development that is provided by the ET-6-6-R is referred to as the ET–social–emotional Quotient (ET-SEQ) throughout the following sections. Regarding the reliability for the total sample across all ages, the scale of social–emotional development has demonstrated a good internal consistency, with a Cronbach’s alpha of 0.75.

### 2.3. Data Analysis

All analyses of the present study were performed using the IBM SPSS Statistics software (Version 27), and an alpha level of 0.05 was set for all statistical tests. First, the total sample of *N* = 93 children aged 3 to 5 years was scanned for outliers and deviations from normality (e.g., skewness and kurtosis). Table 3 depicts descriptive and normality statistics for the index scores VCI, VSI, WMI, and VAI of the WPPSI-IV as well as for the ET-SEQ for the total sample and across sexes. In the total sample, skewness for the WPPSI-IV index scores ranged from −0.45 to 0.01 and was calculated as −0.57 for the ET-SEQ. Kurtosis ranged from −0.95 to 0.57 for the WPPSI-IV index scores and was −0.21 for ET-SEQ. In the female group, skewness ranged from −0.11 to 0.14 for the WPPSI-IV index scores and was calculated as −0.61 for ET-SEQ, whereas kurtosis ranged from −1.04 to −0.46 for the WPPSI-IV index scores and was calculated as −0.59 for ET-SEQ. In the male group, skewness for the WPPSI-IV index scores ranged from −0.88 to 0.33 and was calculated as −0.54 for the ET-SEQ. Kurtosis ranged from −0.86 to 1.46 for the WPPSI-IV index scores and was 0.03 for ET-SEQ. Multivariate kurtosis values of 3.71 (total sample), −2.66 (female group), and 0.08 (male group) were additionally determined. In summary, univariate skewness and kurtosis values as well as multivariate kurtosis values for the total sample did not reveal excessive deviation from normality. Prior to subsequent analyses, the categorical variable parental educational level was dichotomized into category 1 = “low to medium educational level”, combining original categories 1 = “low educational level” and 2 = “medium educational level”, and category 2 = “high to highest educational level”, combining original categories 3 = “high educational level” and 4 = “highest educational level.

Bivariate Pearson correlation coefficients were computed in order to examine the associations among all variables of interest, including the WPPSI-IV primary and ancillary indices *Verbal Comprehension* (VCI), *Visual Spatial* (VSI), *Working Memory* (WMI), and *Vocabulary Acquisition* (VAI) as measures of cognitive skills as well as the ET-social–emotional Quotient (ET-SEQ) as an estimate of social–emotional skills. Since moderation effects were hypothesized for children’s sex and parental educational level, either Spearman or point-biserial correlation coefficients for those variables were also included in the correlational analysis.

A multiple regression analysis (MR 1) was then conducted to test for the hypothesized main effects of cognitive skills on social–emotional skills. Predictor variables in MR 1 included the WPSSI-IV index scores for VCI, VSI, WMI, and VAI, whereas the ET-SEQ was entered as the criterion variable into the regression model. In a second multiple regression analysis (MR 2), the main effects of children’s sex and parental educational level on social–emotional skills were evaluated by adding both predictor variables to the regression model of MR 1. As it is the recommended method for testing for moderation effects [33], a moderated multiple regression analysis (MMR) was finally conducted in order to investigate whether possible effects of cognitive skills on social–emotional skills were moderated by children’s sex and parental educational level. The interaction terms ‘Sex × VCI’ and ‘Sex × VAI’ were thus additionally included in the regression model of MMR to evaluate the hypothesized two-way interactions between children’s sex and *Verbal Comprehension* as well as between children’s sex and *Vocabulary Acquisition*. Furthermore, another four interaction terms were also entered into the regression model of MMR. These interaction terms included two-way interactions between parental educational level and *Verbal Comprehension* (‘Ped × VCI’), parental educational level and *Visual Spatial* (‘Ped × VSI’), parental educational level and *Working Memory* (‘Ped × WMI’), and parental educational level and *Vocabulary Acquisition* (‘Ped × VAI’). Prior to computing product terms and including the according interaction terms in MMR, the respective predictor variables were grand mean-centred in order to reduce the risk of multicollinearity and improve the interpretability of regression coefficients [34]. The overall fit of each of the regression models (MR 1, MR 2, and MMR) was evaluated by assessing *R*^2^, adjusted *R*^2^, and the *F*- and *p*-values for the statistical significance of *R*^2^. Finally, standardized regression coefficients (β) as well as the according *t*- and *p*-values, were assessed for each predictor and moderator variable to check for statistical significance.

In cases of significant interaction effects in the regression model of MMR, the Hayes PROCESS v.4.0 macro [35] was considered for additional moderation analyses to evaluate the relationship between the respective predictor variables and the criterion variable at different levels of the respective moderator variable. This would then allow for the assessment of directionality of single moderation effects, as conditional effects of moderator variables could be visualized for ease of interpretation [36] and confirmed with simple slope tests.

An a priori power analysis that was based on a medium effect size, an alpha error of α = 0.05, a statistical power of 1 − β = 0.80, and four predictor variables suggested a required total sample size of *N* = 85 in MR 1. For the regression analysis in MR 2 including six predictor variables with the other input parameters being the same as in MR 1, the power analysis indicated a required total sample size of *N* = 98. For the third regression analysis including 12 predictor variables in MMR, the power analysis indicated a required total sample size of *N* = 127 for a medium effect size, an alpha error of α = 0.05, and a statistical power of 1 − β = 0.80.

## 3. Results

### 3.1. Descriptive Statistics

Analyses of descriptive characteristics revealed an average index score for WMI in the total sample compared to the normative data (*M* = 99.57, *SD* = 13.35). However, index scores for VCI (*M* = 106.61, *SD* = 12.52), VSI (*M* = 105.69, *SD* = 12.35), and VAI (*M* = 105.44, *SD* = 12.06) appeared to be slightly higher, as would be expected by the according population means. As shown in Table 3, test statistics indicated sex differences of moderate effect size, favouring females in WMI (*t* = −2.53, *p* = 0.01, *d* = −0.53) and ET-SEQ (*t* = −2.02, *p* = 0.05, *d* = −0.42).

### 3.2. Bivariate Correlations

A summary of the bivariate correlations among all relevant variables is presented in Table 4. Correlation coefficients indicated significant small to moderate positive associations between VCI and VSI (*r* = 0.23, *p* < 0.050), VCI and WMI (*r* = 0.21, *p* < 0.05), VCI and VAI (*r* = 0.65, *p* < 0.01), VSI and WMI (*r* = 0.39, *p* < 0.01), VSI and VAI (*r* = 0.28, *p* < 0.01), VSI and ET-SEQ (*r* = 0.25, *p* < 0.05), WMI and VAI (*r* = 0.27, *p* < 0.01), and between VAI and ET-SEQ (*r* = 0.28, *p* < 0.01). The results also revealed small but significant associations between WMI and Sex (*r* = 0.27, *p* < 0.01), ET-SEQ and Sex (*r* = 0.21, *p* < 0.05), VCI and Ped (*r* = 0.23, *p* < 0.05), VSI and Ped (*r* = 0.25, *p* < 0.05), WMI and Ped (*r* = 0.21, *p* < 0.05), and between VAI and Ped (*r* = 0.27, *p* < 0.01).

### 3.3. Regression Analyses

Results of both multiple regression analyses (MR 1 and MR 2) as well as the moderated multiple regression analysis MMR are presented in Table 5. In MR 1, the included predictor variables VCI, VSI, WMI, and VAI explained 12.2% of the variance in the criterion variable ET-SEQ (*R*^2^ = 0.122, *F (R*^2^*) =* 3.059, *p* = 0.02). Moreover, a significant main effect of VSI on ET-SEQ (β = 0.23, *p* = 0.04) indicated a small effect of *Visual Spatial* on social–emotional skills. In MR 2, results showed that 17.4% of the variance in the criterion variable ET-SEQ were explained by the included predictor variables VCI, VSI, WMI, VAI, Sex, and Ped (*R*^2^ = 0.174, *F (R*^2^*) =* 3.023, *p* = 0.01). In addition to a significant effect of VSI on ET-SEQ (β = 0.24, *p* = 0.04), the results also indicated a significant effect of sex on ET-SEQ (β = 0.23, *p* = 0.03), indicating that both *Visual Spatial* and sex had small effects on social–emotional skills. As suggested by the nonsignificant coefficient of determination for the moderated multiple regression analysis MMR, the included predictor variables did not significantly explain variance in the criterion variable ET-SEQ (*R*^2^ = 0.211, *F (R*^2^*) =* 1.783, *p* = 0.07). If this would be the case, however, the results would again support the significant main effects of VSI (β = 0.28, *p* = 0.02) and sex (β = 0.22, *p* = 0.04) on ET-SEQ. Furthermore, this model would have not indicated any salient interaction effects on social–emotional skills. Since the overall model MMR did not provide significant predictive power and did not give any indication of possible moderation effects, MR 2 was selected to be displayed in Figure 1, including all standardized regression coefficient for the predictor variables but excluding any interaction terms from MMR.

## 4. Discussion

The major aim of the present study was to investigate whether cognitive skills such as *Verbal Comprehension*, *Visual Spatial*, *Working Memory*, and *Vocabulary Acquisition* measured by the WPPSI-IV as well as sociodemographic variables such as children’s sex and parental educational level may have a significant effect on social–emotional skills in children aged 3 to 5 years. Furthermore, a second aim of the present study was to examine whether children’s sex and parental educational level might play a moderating role for possible effects of different cognitive skills on social–emotional skills.

Prior to analysing possible effects on social–emotional skills, bivariate correlations among the WPPSI-IV index scores, children’s sex, parental educational level, and the ET-SEQ were first computed to identify significant associations among all variables of interest. Significant correlations were found between the ET-SEQ and VSI as well as between the ET-SEQ and VAI. However, the results of the subsequent multiple regression analyses indicated no significant effects of verbal cognitive skills measured by VCI and VAI on social–emotional skills measured by the ET-SEQ. These findings could be at least partially explained by the fact that those subtests used to compute VCI (*Information* and *Similarities*) are generally not suggested to exclusively measure language skills [37] but also other cognitive domains. Thus, crystallized intelligence, long-term memory, the ability to retain and retrieve knowledge from the environment, verbal perception, comprehension, and expression may also contribute to performances on the subtest *Information*. Performances on the subtest *Similarities*, on the other hand, may strongly depend on verbal concept formation, abstract reasoning, crystallized intelligence, auditory comprehension, memory, associative and categorical thinking, distinction between nonessential and essential features, and verbal expression. It may, therefore, be assumed that VCI also represents a measure of cognitive skills that is not saliently associated with social–emotional skills such as abstract reasoning. Since giving one-word answers in the subtests *Information* and *Similarities* is often sufficient to achieve maximum subtest scores, it can also be suggested that high levels of language competence are not necessarily required in order to achieve high index scores for VCI.

A similar explanation might also be a possible explanation for the absence of a significant effect of VAI on social–emotional skills in the present study. VAI as an ancillary index estimates the level of developmental vocabulary acquisition and is primarily based on expressive and receptive language skills. The underlying subtests for VAI (*Receptive Vocabulary* and *Picture Naming*) are both designed to measure a variety of further cognitive skills [38]. Performances on *Receptive Vocabulary*, on the one hand, also measure receptive language skills, vocabulary development, lexical knowledge, fund of information, long-term memory, and perception of meaningful stimuli, whereas *Picture Naming* is also a measure of expressive language skills, specifically in the area of semantic (i.e., word knowledge) development, knowledge, the fund of information, long-term memory, and the perception of meaningful stimuli. Even though VAI is suggested to operationalize language skills more accurately than VCI does, children either have to select the response option that best represents the word the examiner reads aloud (*Receptive Vocabulary*) or have to name depicted objects (*Picture Naming*). Accordingly, children do not necessarily need to use comprehensive language skills, such as expressive vocabulary and high levels of verbal, grammatical, or syntactic skills, in order to achieve maximum scores in these subtests.

It has to be noted that the findings of Holdnack et al. [14] did not indicate any significant differences in VCI scores between subjects of the Autism spectrum group and a control group. Members of the Autism spectrum group could only be characterized by significantly lower mean scores in the subtest *Comprehension* of the WAIS-IV when compared to members of the control group. In order to give the correct answers on items of the subtest *Comprehension,* which is also included as part of the WPPSI-IV version for the age group 4:0–7:7, participants are required to possess substantial language skills as well as a basic understanding of general principles and social situations.

Although the results of the multiple regression analyses in the present study indicated a significant main effect of VSI on social–emotional skills, it is likely that this effect is not moderated by the parental educational level. The effect of VSI on social–emotional skills is in line with the findings of different studies, which also used the subtest *Block Design* from a Wechsler scale to operationalize the interpretation and organization of visually perceived material [20]. Lemerise and Arsenio [39] proposed a model explaining how visual–spatial processing may affect the processing of social–emotional information. The authors stated that a child’s (visual) perceptual skills are important for detecting and interpreting social and emotional cues from other individuals. In conclusion, their model implies that children with deficits in social–emotional skills may be disadvantaged in socially interacting with others due to their difficulties in detecting specific emotional states of others. The ability to gather such information, however, is indispensable for guiding and adapting behaviour in social contexts.

Regarding structural validity, the findings of the present study also tend to strongly support the new second-order five-factor model of the WPPSI-IV for the age group 4:0–7:7 compared to the three-factor structure of the former version [40]. Based on factor analyses, the former index *Performance-IQ* of the Wechsler Preschool and Primary Scale of Intelligence—Third Edition (WPPSI-III; [41]) was divided into separate indices for *Visual Spatial* and *Fluid Reasoning* in the WPPSI-IV. Since *Visual Spatial* was found to be a potent predictor for social–emotional skills in the present study, the new second-order five-factor structure of the WPPSI-IV may thus provide a more in-depth analysis of specific cognitive deficits that might otherwise be neglected.

Given that working memory capacity is involved in taking another individual’s perspective when communicating with them [18], current research also suggests an important role of working memory for social–emotional skills. Baddeley’s multi-component model of working memory specifies two domain-specific storage systems for temporal storage and rehearsal or manipulation of domain-specific information: the phonological loop for processing verbal information and the visual–spatial sketchpad for processing visual and spatial information [42]. Since auditory stimuli need to be stored and retrieved for social communication, the phonological loop may be more responsible for the association between working memory and social–emotional skills. Findings reported by Holdnack et al. [14] also point to such an association, as the authors could show that participants diagnosed with Asperger’s Syndrome showed lower mean performances on WAIS-IV tasks measuring auditory working memory than controls. The correlation analysis and the multiple regression analyses in the present study, however, both failed to indicate a meaningful effect of WMI on the ET-SEQ. This may be due to the fact that working memory in the WPPSI-IV is measured by the subtests *Picture Memory* and *Zoo Locations* which, in turn, were originally designed to measure visual (*Picture Memory*) and visual–spatial (*Zoo Locations*) rather than auditory working memory.

Regarding the possible effects of socio-demographic variables, regression analyses of the present study showed that children’s sex had a statistically significant effect on social–emotional skills but cannot be suggested to moderate any other effects of cognitive measures. Mean differences also indicated that females had a significant advantage in social–emotional skills over males. Although these findings are in accordance with those in previous studies, it is still unclear whether sex differences are primarily caused by biological or social–cultural factors. It is noteworthy that there is an ongoing controversial debate on this topic in research literature [43,44]. Stowe et al. [24], for instance, suggested that developmental problems of girls at preschool age tend to be less perceivable than those of boys. Although girls and boys may equally be affected by deficits in their social–emotional development (e.g., difficulties in taking another individual’s perspective or detecting and interpreting social and emotional cues from other persons), it is likely that boys tend to show more disruptive and externalizing behaviour, while girls tend to show more internalizing behaviour.

### Limitations

There are some limitations in the present study that need to be addressed and should also be considered when replicating or generalizing the present findings. Even though an a priori power analysis showed that the achieved sample size of *N* = 93 should be sufficient to detect significant moderate effects for the correlational and the multiple regression analyses MR 1 and MR 2 in the present study, the power analysis was based on moderate effects. Furthermore, an a priori power analysis for the moderated multiple regression analysis indicated that the achieved total sample size was not large enough to detect possible moderate main and interaction effects. Based on the sample size, the observed measure of global fit (coefficient of determination), an alpha error probability of 0.05, and the number of predictors, post hoc power analyses for each of the regression models indicated sufficient statistical power for MR 1 with four predictors (*N* = 93, *R*^2^ = 0.12, 1 − β = 0.80), as well as for MR 2 with six predictors (*N* = 93, *R*^2^ = 0.18, 1 − β = 0.93). The actual statistical power for MMR with 12 predictors, however, was deemed inadequate (*N* = 93, *R*^2^ = 0.24, 1 − β = 0.32), indicating insufficient explanatory power for MMR. Since the regression analysis for MMR might thus have failed to detect rather small but relevant main or interaction effects, comprehensive large-sample analyses should be considered in future research on this topic.

Another shortcoming of the present study is highlighted by the fact that previous studies could already show an association between processing speed and social–emotional skills [14]. However, only the WPPSI-IV version for the age group 4:0–7:7 provides a measure of processing speed but only data of a total of *N* = 61 children aged 4 years and older were available for the present analyses. Even though the overall sample of the present study appeared to be mostly representative of the German census, the group of children with parents featuring the highest educational level appeared to be clearly over-represented. Given that the variance in parental educational level was also limited in the present sample, this could explain why the present analyses failed to detect any moderate effects of parental education on social–emotional skills. It may also be that a dichotomized variable for the parental educational level such as the one used in the present study was not sensitive enough to detect such effects. Therefore, future studies should consider using the ISCED (International Standard Classification of Education) guidelines [24] to assess parental educational level more precisely. In general, future research on associations between performances in WPPSI-IV and social–emotional skills should be based on a larger sample size and a more population-representative sample.

As the present study is cross-sectional in nature, inferences about causality are limited by its very definition. While there is a theoretical basis for the regression models including both cognitive and socio-demographic variables as predictors as well as considering moderation effects based on research literature, longitudinal analyses of the hypothesized associations are required to determine causal inferences. Moreover, future longitudinal studies may also provide more adequate analyses of causal effects on social–emotional development over time.

It should also be noted that the WPPSI-IV was the only test battery selected to measure and operationalize cognitive skills in the present study. Since using alternative and more diverse approaches could improve the measurement of cognitive skills by reducing potential methodological biases, future research should focus on multi-methodological measurements of cognitive skills when analysing possible effects on the social–emotional skills of preschoolers.

Another limitation of the present study is the ET-6-6-R, which was used to measure social–emotional skills. Social–emotional development, on the other hand, was assessed using a parent questionnaire. However, this approach only allowed for the assessment of social–emotional skills of children and might not have been sufficient for a comprehensive assessment of social–emotional development in general. It is therefore recommended for future studies to use multiple rather than one questionnaire for the external assessment of social–emotional development (e.g., questionnaires for parents and kindergarten teachers). The majority of research has also emphasized the importance of different sources of factual information especially for childhood emotional syndromes. It may thus be crucial for clinicians to collect information from different sources especially when children are older than nine years [45]. However, the ability of younger children to express themselves and report their own symptoms may be quite limited. Verhulst and Van der Ende [46] also reported that parents are generally familiar with their child’s well-being in many situations and throughout time and can therefore provide valid information about their child’s symptoms.

## 5. Conclusions

Despite the aforementioned limitations, the present study makes further contributions to the literature focusing on the role of cognitive skills, sex, and parental education for social–emotional skills in children aged 3 to 5 years. First of all, visual–spatial skills as measured by the WPPSI-IV can be suggested to have a meaningful effect on social–emotional skills. Moreover, the present findings also indicated that children’s sex might have a reasonable predictive value on social–emotional skills. In general, future research should clarify whether those cognitive skills and socio-demographic characteristics that may partially explain variations in social–emotional skills in early childhood can be considered to have an impact on the development of social–emotional skills and related psychological disorders in later childhood or adolescence.

## Figures and Tables

**Figure 1 children-09-00730-f001:**
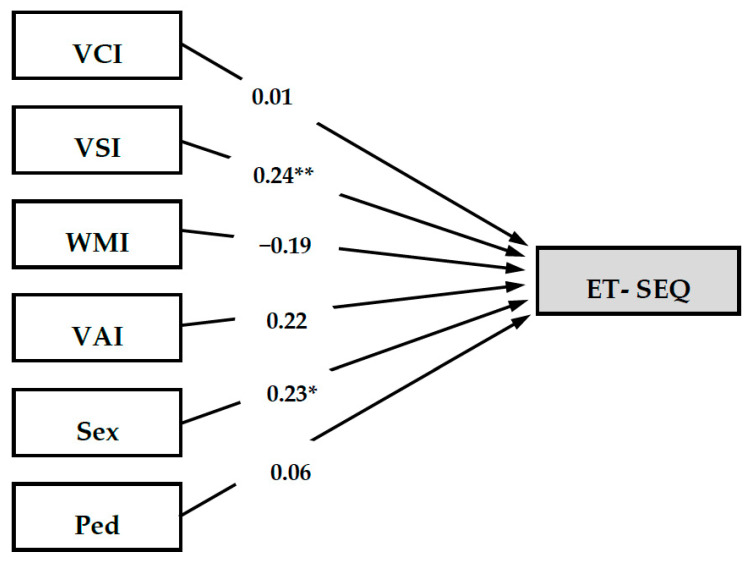
Statistical model for the multiple regression analysis MR 2 including standardized regression coefficients for cognitive skills, sex, and parental educational level on social–emotional skills (*N* = 93). Note. VCI = Verbal Comprehension Index, VSI = Visual Spatial Index, WMI = Working Memory Index, VAI = Vocabulary Acquisition Index, ET-SEQ = Social–Emotional Development Quotient of the ET-6-6-R. Children’s sex is defined as 1 = male, 2 = female. Ped = Parental educational level is defined as the highest level of education achieved by either one parent or both (1 = low to medium educational level, 2 = high to highest educational level). R^2^_adj._ = adjusted R^2^, F (R^2^) = F score for the change in R^2^. * *p* < 0.05, ** *p* < 0.01.

**Table 1 children-09-00730-t001:** Socio-demographic characteristics of the sample according to children’s sex, age, parental education, and migration background (*N* = 93).

Age Group (Months)	Sex	*N*	Parental Education (%)				Mig. (%)
			1	2	3	4	
36–47	Female	17	0	5.9	29.4	64.7	5.9
	Male	15	13.3	26.6	13.3	46.8	20.0
	Total	32	6.3	15.6	21.9	56.2	12.5
48–59	Female	17	11.8	11.8	17.6	58.8	5.9
	Male	12	0	0	33.4	66.6	16.7
	Total	29	6.9	6.9	24.1	62.1	10.3
60–71	Female	13	15.4	15.4	23.1	46.1	23.1
	Male	19	5.3	26.3	5.3	63.1	31.6
	Total	32	9.4	21.9	12.5	56.2	28.1
36–71	Female	47	8.5	10.6	23.4	57.4	10.6
	Male	46	6.5	19.6	15.2	58.7	23.9
	Total	93	7.5	15.1	19.4	58.1	17.2

Note. Parental education is defined as the highest level of education achieved by either one parent or both (1 = low educational level, 2 = medium educational level, 3 = high educational level, 4 = highest educational level). Mig (%) = percentage of cases with migration background (migration background is indicated when either the child or at least one parent was not born in Germany). Data collection and testing: parents and kindergarten teachers were informed about the main goals and procedures of the study as well as about data processing and protection. Parents were then required to give their approval by signing a written informed consent form.

**Table 2 children-09-00730-t002:** Item examples for the assessment of social–emotional development in the Developmental Test for Children Aged 6 Months to 6 Years—Revision (ET-6-6-R) for children aged 60 to 72 months.

Measured Dimension	Example Item
Social interaction with peers	It invites other children over to itself and is happy to be invited. It specifically selects certain children with whom it would like to have contact and plays with them even when adults are absent.
Compliance with social rules and norms	It reliably follows certain household rules (e.g., does not simply climb on tables or jump on upholstered furniture).
Social autonomy	It can easily be separated from you for a few hours if it is cared for by someone he knows well.
Social identity and role adoption	It reacts everyday situations or stories with dolls or play figures.
Emotional regulation	It can regulate its emotions sufficiently itself. It can disappointments, concerns, fears, embarrassment, or anticipation endure. Occasional tantrums still occur.
Social interaction with strangers	It also answers unknown people on the phone.

**Table 3 children-09-00730-t003:** Descriptive and normality statistics for the WPPSI-IV index scores VCI, VSI, WMI, and VAI, as well as the ET-SEQ for the total sample and across sexes including test statistics for sex differences.

Variables	Total (*N* = 93)				Female (*n* = 47)				Male (*n* = 46)				Comparisons		
	*M*	*SD*	Skew	Kurt	*M*	*SD*	Skew	Kurt	*M*	*SD*	Skew	Kurt	Test Statistics		
			*Z*	*Z*			*Z*	*Z*			*Z*	*Z*	*t*	*p*	*d*
VCI	106.61	12.52	−0.20	0.57	107.51	12.26	0.12	−0.46	105.70	12.86	−0.46	1.46	−0.70	0.49	−0.15
VSI	105.69	12.35	0.01	−0.95	106.34	12.20	0.03	−1.04	105.02	12.59	−0.02	−0.86	−0.51	0.61	−0.11
WMI	99.57	13.35	−0.03	−0.22	102.94	10.85	−0.11	−0.46	96.13	14.83	0.33	−0.06	−2.53	0.01	−0.53
VAI	105.44	12.06	−0.45	0.45	105.96	12.14	0.14	−0.32	104.91	12.09	−0.88	1.22	−0.42	0.68	−0.09
ET-SEQ	12.22	2.33	−0.57	−0.21	12.81	2.74	−0.61	−0.59	11.61	2.99	−0.54	0.03	−2.02	0.05	−0.42
Multivariate				3.71				−2.66				0.08			

Note. VCI = Verbal Comprehension Index, VSI = Visual Spatial Index, WMI = Working Memory Index, VAI = Vocabulary Acquisition Index, ET-SEQ = Social–Emotional Development Quotient of the ET-6-6-R. Skew = skewness, Kurt = kurtosis, Multivariate = Mardia’s multivariate kurtosis estimate. Test statistics = independent *t*-tests for mean differences in subtest scores between females and males, *d* = Cohen’s *d*.

**Table 4 children-09-00730-t004:** Bivariate correlations among the WPPSI-IV primary index scores, the ET-SEQ, children’s sex, and parental educational level (*N* = 93).

Variables		1	2	3	4	5	6	7
1.	VCI	1						
2.	VSI	0.23 *	1					
3.	WMI	0.21 *	0.39 **	1				
4.	VAI	0.65 **	0.28 **	0.27 **	1			
5.	ET-SEQ	0.20	0.25 *	0.03	0.28 **	1		
6.	Sex	0.07	0.05	0.27 **	0.04	0.21 *	1	
7.	Ped	0.23 *	0.25 *	0.21 *	0.27 **	0.14	0.08	1

Note. VCI = Verbal Comprehension Index, VSI = Visual Spatial Index, WMI = Working Memory Index, VAI = Vocabulary Acquisition Index, ET-SEQ = Social–Emotional Development Quotient of the ET-6-6-R. Sex is defined as 1 = male, 2 = female. Ped = Parental educational level is defined as the highest level of education achieved by either one parent or both (1 = low to medium educational level, 2 = high to highest educational level). * *p* < 0.05, ** *p* < 0.01.

**Table 5 children-09-00730-t005:** Model summary for both multiple regression analyses MR 1 and MR 2 and the moderated multiple regression analysis MMR including main and interaction effects on social–emotional skills (*N* = 93).

Predictor Variables	Model MR 1					Model MR 2					Model MMR				
	*B*	*SE*	β	Test Statistics		*B*	*SE*	β	Test Statistics		*B*	*SE*	β	Test Statistics	
				*t*	*p*				*t*	*p*				*t*	*p*
VCI	0.01	0.03	0.03	0.25	0.81	0.01	0.03	0.01	0.09	0.93	0.01	0.03	0.02	0.011	0.91
VSI	0.06	0.03	0.23	2.09	0.04	0.06	0.03	0.24	2.15	0.04	0.07	0.03	0.28	2.37	0.02
WMI	−0.03	0.02	−0.12	−1.11	0.27	−0.04	0.02	−0.19	−1.71	0.09	−0.05	0.03	−0.22	−1.83	0.07
VAI	0.05	0.03	0.22	1.65	0.10	0.06	0.03	0.22	1.71	0.09	0.05	0.03	0.21	1.55	0.13
Sex	-	-	-	-	-	1.32	0.59	0.23	2.25	0.03	1.30	0.61	0.22	2.12	0.04
Ped	-	-	-	-	-	0.39	0.73	0.06	0.54	0.59	0.04	0.81	0.01	0.06	0.96
Sex × VCI	-	-	-	-	-	-	-	-	-	-	0.02	0.06	0.05	0.37	0.71
Sex × VAI	-	-	-	-	-	-	-	-	-	-	−0.09	0.07	−0.18	−1.30	0.24
Ped × VCI	-	-	-	-	-	-	-	-	-	-	−0.01	0.07	−0.01	−0.08	0.94
Ped × VSI	-	-	-	-	-	-	-	-	-	-	−0.03	0.07	−0.01	−0.51	0.61
Ped × WMI	-	-	-	-	-	-	-	-	-	-	0.02	0.06	0.05	0.37	0.71
Ped × VAI	-	-	-	-	-	-	-	-	-	-	−0.05	0.07	−0.09	−0.64	0.52
*R*	0.35					0.42					0.46				
*R*^2^ (*R*^2^_adj._)	0.12 (0.08)					0.17 (0.12)					0.21 (0.09)				
*F (R* ^2^ *)*	3.06 *					3.02 **					1.78				

Note. VCI = Verbal Comprehension Index, VSI = Visual Spatial Index, WMI = Working Memory Index, VAI = Vocabulary Acquisition Index. Sex is defined as 1 = male, 2 = female. Ped = Parental educational level is defined as the highest level of education achieved by either one parent or both (1 = low to medium educational level, 2 = high to highest educational level). All predictor variables were grand mean-centred in MMR. Test statistics = two-tailed *t*-tests for statistical significance of the standardized regression coefficient ß. *R*^2^_adj._ = adjusted *R*^2^, *F (R*^2^*) = F* score for the change in *R*^2^. * *p* < 0.05, ** *p* < 0.01.

## Data Availability

Since the research is based on health service research (Versorgungsforschung) the data are not publicly available.

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
