# Peer review of "The Role of Cognitive Skills, Sex, and Parental Education for Social–Emotional Skills: A Cross-Sectional Study on the WPPSI-IV Performances of Children Aged 3 to 5 Years"

_children, 2022, doi:10.3390/children9050730_

Round 1
Reviewer 1 Report
This paper presents a quantitative study how a variety of cognitive predictors related to socio-emotional skills, and the possible moderating role of contextual factors for this relationship, in young children. This paper aimed to add to the research by exploring the associations between typically independent processes, that of cognitive and socio-emotional development, through the use of well-established test batteries. However, I do have concerns about the overall power of the study for the analysis conducted – and ultimately this impacts the contribution the study can have to the field.
- I would like to see a little more initial introduction to both cognitive and socio-emotional skills – both are very big and all-encompassing terms (and recognition of this). How are the authors defining these, and this could lead to the support of the constructs being measured and using the chosen measures.
- It would be good to see power calculations to support the analysis that was conducted e.g., is a multiple regression with 12 predictor variables powered with only 93 children. My concern that is the analysis may be under powered. This was not reported until the limitations, and noted that the study was only powered for moderate affects. As the authors note this is a considerable limitation impacting the conclusions that can be drawn.
- I am right in thinking both sex and parent education have been entered into the regression as continuous variables, parent education as a four category response should really be treated as categorical, especially considering the skew towards the higher categories. It would also then be interesting to see if the effects of parent education reach a limit at high, and whether higher education offers any extra “advantage”.
- A major limitation/issue was that the socio-emotional skill component was measured via a parent questionnaire, while the cognitive skills were performance-based assessments. This will likely have a big impact on the associations between these skills. While this was mentioned it warrants a deeper discussion of relationships between ability/performance aged skills and self-reported - as often weak correlations are found between these different sorts of measures.
Reviewer 2 Report
Thank you very much for giving me the opportunity to review the manuscript entitled: “The role of cognitive skills, sex, and parental education for social-emotional skills: A cross-sectional study on the WPPSI-IV performances of children aged 3 to 5 years”. It aims to analyse the effects of cognitive skills measured by the WPPSI-IV as a predictor of social-emotional skills in children, and to examine whether children's sex and parental education may play a moderating role in the possible effects of cognitive skills on emotional variables.
I would like to congratulate the authors on their work, I found it an interesting article, with enough content for anyone to understand without having to be an expert on the subject. The authors present us with an interesting study, as their work aims to generate content related to a gap in the existing research on the relationship between cognitive skills and social-emotional development in children aged 3 to 5 years. In addition, they delve into the possible variation in cognitive skills outcomes by gender and parental education level.
The analyses seem to me to be adequate, and a strength of this research is the age of the participants. Few studies are found in such young children, taking into account that data collection in this group is often complicated.
I have nothing to add, I will only comment on small things that can be modified during the editing process as they are formatting issues.
Abstract: In the background there is no introduction, it goes directly to the aims. Either put a short introduction or switch to aims.
Check the foot of table 1, it is displaced.
I noticed that in table 3 you have the decimal values with zero and in table 4 and 5 you omit it. Please be consistent in this, as well as in the number of decimals.
Round 2
Reviewer 1 Report
The authors have added brief definitions purporting to how they are conceptualising socio-emotional skills and cognitive skills. But I still have concerns that these are all encompassing constructs for which there exists much debate on the make-up of these skills that is not acknowledged. Further, the suggestion that “little is still known about the associations among these domains” is not quite accurate. There is a vast body of work that has drilled down into these constructs to pinpoint how various different aspects of socio-emotional skills and cognitive skills are related. For example, the body of work exploring connections between executive function skills such as inhibition and working memory on emotional understanding and regulation, and theory of mind in children aged 3-5 years of age. Robust associations between these parts of socio-emotional and cognitive skills is much reported and debated. This history of research is not reflected in the current paper, and thus the rationale for why we need to understand how the constructs measure in the current study add to the literature base.
The section on cognitive skills definition helps support the approach to measuring this and the factors focused on with this age group. However, the definition of social emotional skills cites five dimensions: (1) social competence; (2) attachment; (3) emotional competence; (4) self-perceived competence; and (5) temperament/personality. Yet what is measured is (1) Social interaction with peers, (2) Compliance with social rules and norms, (3) Social autonomy, (4) Emotional regulation, and (5) Social interaction with strangers. Why is then that these aspects of socio-emotional skills are measured and analysed, how do they fit with the definition and theory on socio-emotional skills.
My main concern was whether the analysis conducted was powered, and the authors also raise this as a limitation. It is good to see this is a now added, but this does not alleviate my concerns that the study is underpowered for the analysis conducted. Data has been collected for 93 children, and with a regression model with 12 predictors accounting for the interaction terms is underpowered. The authors state 127 children would be needed, this is a lenient suggestion, some statisticians would argue many more for a model of this number. It would be more useful to carry these calculations out post hoc and report how well powered each analysis actually was, so how under powered was MMR with 12 predictors?
I am still confused about how the categorical variable parent education has been added to the model MR2 and MMR. Has it been added as a continuous variable 1-4, or as a categorical variable 1 = low educational level, 2 = medium educational level, 3 = high educational level, 4 = highest educational level. This is needed to aid the understanding and interpretation of the models. If the former, then this assumes a linear relationship (is this predicted/expected), but is problematic as the variable is heavily skewed towards the upper ends. If the latter, while this adds detail and better represents the variable, this needs to be added as dummy variables of each category which in turn ups the number of predictors and thus the number of participants needed – further underpowering the analyses. One way to help this would be to dichotomize the variable, but as the authors note this would loose detail.
Round 3
Reviewer 1 Report
The authors have elaborated on definitions conceptualising socio-emotional skills. But this still does not quite bridge the gap theory and measurement. E.g., between Denham‘s distinction of 5 domains: (1) social competence; (2) attachment; (3) emotional competence; (4) self-perceived competence; and (5) temperament/personality) and their measurement of a different 5 domains (1) Social interaction with peers, (2) Compliance with social rules and norms, (3) Social autonomy, (4) Emotional regulation, and (5) Social interaction with strangers. The authors suggest that there are important milestones to social development (e.g., peer interactions, managing emotions, building friendships etc.) as well as for emotional development (e.g., emotion regulation). Thus it would be useful for the authors add a specific statement, that it is a culmination of these milestones they are to focus on and why, and support that this links to the measurement within the current paper.
The authors have made significant acknowledgement that the study is underpowered, and this limits the conclusions that can be made.
Clarity on the on how parental education is measured, quantified and added to the model is now offered. This eliminates issues with skew to higher parent education on this variable if treated as continuous data.
